



# Impacts on cloud radiative effects induced by coexisting aerosols converted from international shipping and maritime DMS emissions

Qinjian Jin[1], Benjamin S. Grandey[2], Daniel Rothenberg[1], Alexander Avramov[1†], Chien Wang[1,2]

[1] Center for Global Change Science, Massachusetts Institute of Technology, Cambridge, Massachusetts, USA.
[2] Center for Environmental Sensing and Modelling, Singapore–MIT Alliance for Research and Technology, Singapore.
† Now at Department of Environmental Science, Emory University, Atlanta, Georgia, USA.

*Correspondence to*: Qinjian Jin (jqj@mit.edu)

**Abstract.** International shipping emissions (ISE), particularly sulfur dioxide, can influence the global radiation budget by interacting with clouds and radiation after being oxidized into sulfate aerosols. A better understanding of the uncertainties in estimating the cloud radiative effect (CRE) of ISE is of great importance in climate science. Many international shipping tracks cover oceans with substantial natural dimethyl sulfide (DMS) emissions. The interplay between these two major aerosol sources on cloud radiative effects over vast oceanic regions with relatively low aerosol concentration is an intriguing yet poorly addressed issue confounding estimation of the cloud radiative effects of ISE. Using an Earth system model including two aerosol modules with different aerosol mixing configurations, we derive a significant global net CRE of ISE ($-0.153$ W m$^{-2}$ with $p$=0.01 and standard error of 0.004 W m$^{-2}$) when using emissions consistent with current ship emission regulations. This global net CRE would become much weaker and actually insignificant ($-0.001$ W m$^{-2}$ with $p$=0.98 and standard error of 0.007 W m$^{-2}$) if a more stringent regulation were adopted. We then reveal that the ISE-induced CRE would achieve a significant enhancement when lower DMS emission is prescribed in the simulations, owing to the sub-linear relationship between aerosol concentration and cloud response. In addition, this study also demonstrates that the representation of certain aerosol processes, such as mixing states, can influence the magnitude and pattern of the ISE-induced CRE. These findings suggest a re-evaluation of the ISE-induced CRE with consideration of DMS variability.

## 1 Introduction

Marine stratiform clouds have a strong cooling effect on the climate system. They cover about 30% of the global ocean surface (Warren et al., 1988), and can reflect more solar radiation back to space than the dark ocean surface at cloud-free conditions. On the other hand, low-altitude marine stratiform clouds form and develop near to the ocean surface (only several degrees cooler than ocean surface) and thus have limited impacts on the longwave radiation balance (Klein and Hartmann, 1993). Therefore, the annual-mean net radiative effect of cloud at the top of the atmosphere (TOA) is negative (i.e., cooling) and can be up to $-20$ W m$^{-2}$ on the global scale (Boucher et al., 2013). Consequently, even a few percent change in marine stratocumulus cloud cover can double or offset the anthropogenic global warming due to greenhouse gases.

Sulfate aerosols are efficient cloud condensation nuclei and control the formation of marine clouds and their micro- and macro-physical properties (McCoy et al., 2015). The international shipping-emitted sulfur dioxide from combustion of heavy fossil oil (Figure 1) can be oxidized to sulfate aerosols that can increase cloud droplet number concentrations, cloud liquid water path, and planetary albedo, resulting in more solar radiation being reflected back to space, exerting a cooling effect on the climate system (Capaldo et al., 1999; Devasthale et al., 2006; Lauer et al., 2007; Lauer et al., 2009). Although international shipping emissions (ISE) contribute only about 5% (5.6 Tg S yr$^{-1}$) to the total anthropogenic sulfur emissions (Corbett and Koehler, 2003; Endresen et al., 2005; Klimont et al., 2013), they dominate the sulfur concentration across much of the ocean,





such as the North Pacific Ocean (NPO) and the North Atlantic Ocean (NAO), as shown in Figure 2. Therefore, the radiative
impact of the ISE via perturbing marine stratocumulus clouds could be large—especially because marine stratocumulus clouds
are often collocated with busy shipping lanes (Neubauer et al., 2014).
Nevertheless, the estimated global annual mean of the ISE-induced net cloud radiative effect (CRE) at TOA has large
uncertainties due to the complication in simulating clouds and aerosol–cloud interactions, ranging from −0.60 to −0.07 W m$^{-2}$
(Capaldo et al., 1999; Lauer et al., 2007; Eyring et al., 2010; Righi et al., 2011; Peters et al., 2012; Partanen et al., 2013).
Compared with net CRE, the net direct radiative effect (DRE) of shipping emissions is much weaker, with a magnitude of only
−0.08 to −0.01 W m$^{-2}$, approximately one tenth of the former (Endresen et al., 2003; Schreier et al., 2007; Eyring et al., 2010).
Besides shipping-emitted sulfur compounds, oceanic phytoplankton-derived dimethyl sulfide (DMS) is another
significant component in the atmospheric sulfur cycle over oceans (Grandey and Wang, 2015; Mahajan et al., 2015; McCoy et
al., 2015; Tesdal et al., 2016). DMS can be oxidized by hydroxyl radical or nitrate radical to produce sulfur dioxide and finally
converted to sulfate aerosols (Boucher et al., 2003). The global total DMS emission is estimated to range from 8 to 51 Tg S yr$^{-1}$
based on model simulation (Quinn et al., 1993; Dentener et al., 2006); this uncertainty range is itself substantially larger than the
total sulfur emissions from shipping. The global annual mean of the DMS-induced net CRE at TOA is from −2.03 to −1.49 W
m$^{-2}$ determined by DMS climatology (Gunson et al., 2006; Thomas et al., 2010; Mahajan et al., 2015).
Most of the aforementioned studies addressed separately the impacts on CRE of shipping and DMS emissions, largely
ignoring the potential nonlinearity in the response of cloud radiative effects to aerosol variations when these sulfate aerosols
from two different sources often collocate in the relatively clean marine atmosphere, such as NPO and NAO (Figure 1). The
nonlinearity between DMS emission and the associated CRE was studied previously without taking into account the shipping
emissions (Pandis et al., 1994; Russell et al., 1994; Gunson et al., 2006; Thomas et al., 2011). Here, to evaluate the CRE induced
by both ISE and DMS emissions with a consideration of their interactions, we selected three regions for detailed analysis: the
NPO and NAO where the ISE dominate the concentrations of sulfur dioxide and sulfate aerosols, and the Southern Ocean where
DMS is the dominant source (Figure 2).
This study employs an Earth system model including an interactive aerosol model that simultaneously resolves both
external and internal mixtures of sulfate, black carbon, and organic carbon aerosols. Aerosol mixing in this way can resolve
aerosol activation process more realistically than either mixing all aerosol species internally or ignoring any mixing at all. By
comparing the results with the default aerosol scheme that ignores above mixing processes, we also quantify the impacts of
various assumptions of aerosol mixing states on estimates of the CRE of ISE and DMS emissions. We further quantify the ISE-
induced CRE based on various regulations of the International Maritime Organization (IMO) on the fuel sulfur content.
Therefore, our findings have important implications for policy makers and future estimates of CRE induced by both ISE and
DMS emissions.
**2 Methods**
**2.1 Climate model**
The Community Earth System Model version 1.2.2 (CESM1.2.2) is configured with the Community Atmosphere Model
version 5.3 (CAM5.3). CAM5.3 includes a modal aerosol model with an option of 3 or 7 lognormal distributions of aerosol size
(MAM3 or MAM7). In this study, a new modal aerosol model—the two-Moment, Multi-Modal, Mixing-state resolving Aerosol
model for Research of Climate (MARC; version 1.0.3 here) (Kim et al., 2008; Kim et al., 2014; Rothenberg and Wang, 2016;
Rothenberg et al., 2017; Rothenberg and Wang, 2017; Grandey et al., 2018) is introduced and used to diagnose both the DRE



and CRE of ISE. The details of MARC are described in the following section. Aerosol DRE are represented by coupling between
aerosols and radiation. Aerosol CRE are included by activating aerosols to work as cloud condensation nuclei and ice nuclei in
the stratiform clouds (Morrison and Gettelman, 2008; Gettelman et al., 2010). Parameterization of aerosol activation is based on
particle size and hygroscopicity of aerosols. Similar to other climate models, CAM5 does not include aerosol's influence on
convective clouds.

**2.2 MAM3**

The default aerosol scheme in CESM 1.2.2, MAM3, has three modes, each with a lognormal size distribution: Aitken,
accumulation, and coarse. Various aerosol species are internally mixed within each mode. Aitken mode is a mixture of sulfate,
secondary organic carbon and sea salt; accumulation mode is mixture of sulfate, black carbon, primary organic carbon,
secondary organic carbon, dust, and sea salt; coarse mode is a mixture of dust, sea salt, and sulfate (Liu et al., 2012).

**2.3 MARC**

MARC uses seven modes with different lognormal size distribution to represent the population of sulfate and
carbonaceous aerosols: three modes for sulfate (nucleation or NUC, Aitken or AIT, and accumulation or ACC), one each for
pure black carbon (BC) and pure organic carbon (OC), one mixture of BC–sulfate in core-shell structure (MBS), and one mixture
of OC–sulfate (internal mixture; MOS). MARC predicts total particle mass and number concentrations while assuming the
standard deviation within each of the seven modes to define at any given time the lognormal distribution of particle size. In
addition, carbonaceous mass concentrations inside MBS and MOS are also predicted to allow the mass ratios between sulfate
and carbonaceous compositions evolve over time, changing the optical and chemical properties of the mixed aerosols. The
emissions of mineral dust and sea salt that MARC uses are calculated by the land surface model and atmosphere model,
respectively (Mahowald et al., 2006; Albani et al., 2014; Scanza et al., 2015). Mineral dust and sea salt are each represented by
four bins with fixed sizes in MARC. For details of MARC aerosol mode size distribution and chemical parameters, please refer
to Rothenberg et al. (2017).

**2.4 Radiation diagnostics**

In the diagnostic mode of CESM-MARC, the DRE are diagnosed by calling the radiation scheme three times in each
radiation time step. The first call does not include any aerosols. The second call includes only mineral dust and sea salt aerosols.
The third call includes all aerosols. The first and third call are diagnostic while the second call is prognostic. Therefore, DRE of
only dust and sea salt aerosols are prognostic while all other aerosols including ISE are diagnostic. In the complimentary MAM3
simulations, the first radiation call (prognostic) includes all aerosols while the second call (diagnostic) excludes all aerosols. In
this way, the DRE and CRE of ISE can be isolated and evaluated separately. Note that all radiative effects are calculated at TOA.

**2.5 Experimental design**

Three groups of simulations are designed to evaluate: (*a*) the DRE and CRE of ISE, and (*b*) the sensitivity of the ISE-
induced CRE to both DMS emissions and aerosol mixing assumptions. CAM5 was run at a horizontal resolution of 1.875°×2.5°
and 30 vertical layers with sea surface temperature (SST), sea ice, greenhouse gas concentrations prescribed at the level of year
2000. The aerosol emissions in year-2000 were used except for modified shipping and DMS emissions. The DMS emission is
prescribed with a global annual average of 18.2 Tg S yr$^{-1}$ in DMS reference simulations (Dentener et al., 2006). Each simulation
runs for 32 years driven by 12-month cyclic climatological sea surface temperature, with the first 2 years discarded as spin up.





Since the observed SST was used, a 2-year period of spin up should be enough for aerosol concentrations and other model
components to reach an equilibrium state (e.g., Righi et al., 2011).

The first group uses CESM-MARC and includes four simulations, which share the same DMS reference emissions

(*DMSRef*) while differing in four various ISE of sulfur compounds (sulfur dioxide and sulfate) (Table 1). *ShipZero_DMSRef*
simulation is integrated excluding all aerosol and aerosol precursor emissions from ISE, i.e., sulfur dioxide, sulfate aerosol,
organic carbon aerosol, and black carbon aerosol. The other three simulations—*ShipLow_DMSRef*, *ShipRef_DMSRef*, and
*ShipHigh_DMSRef*—include the standard emissions of carbonaceous aerosols (e.g., BC and OC) while use three various
emission scenarios of sulfur compounds from ISE. The three emission scenarios are based on the assumptions of sulfur content
of the heavy fuel oils for ocean-going ships. Currently, the average sulfur content is 2.7% (Corbett and Koehler, 2003; Endresen
et al., 2005), which is equivalent to about 5.6 Tg S year$^{-1}$, referred to as *ShipRef*. On the other hand, as of 2013 the high sulfur
fuel oil that has 3.5% sulfur content continued to be permitted outside the Emission Control Areas (Lauer et al., 2009; Winebrake
et al., 2009), referred to as *ShipHigh*. However, the IMO has planned to lower the sulfur content to 0.5% outside the Emission
Control Areas (Corbett et al., 2007; Winebrake et al., 2009; Notteboom, 2010; International Maritime Organization, 2016) after
2020, referred to as *ShipLow*. In *ShipLow* and *ShipHigh*, the total global sulfur shipping emissions are 1.0 and 7.2 Tg S year$^{-1}$,
respectively. The differences between these three and the zero shipping emission scenarios represent how various regulations on
marine fuel influence the ISE-induced CRE.

The second group also uses CESM-MARC and is comprised of three pairs of simulations: (*ShipRef_DMSZero*,

*ShipZero_DMSZero*), (*ShipRef_DMSLow*, *ShipZero_DMSLow*), and (*ShipRef_DMSRef*, *ShipZero_DMSRef*). The annual
emission of DMS is 18.2 Tg S year$^{-1}$ in the *DMSRef* simulations (Dentener et al., 2006; Liu et al., 2012), and half of that in the
*DMSLow* simulations. DMS emission is excluded in the *DMSZero* simulations. Each pair of the simulations include *ShipZero*
and *ShipRef*, the difference of which represents the ISE-induced impacts. The purposes of the *DMSZero* and *DMSLow*
simulations are to quantify the sensitivities of the ISE-induced CRE (i.e., the difference of CRE in each pair of simulations) to
DMS emission and the associated large uncertainty in DMS emission, respectively (Quinn et al., 1993; Dentener et al., 2006).
DMS emission in the *DMSLow* simulation is 9.1 Tg S year$^{-1}$, which is close to the lower boundary of DMS emission estimates,
i.e., 8 Tg S year$^{-1}$ (Quinn et al., 1993). Such sensitivities are examined by calculating the differences among the three pairs of
DMS simulations.

The third group is the same as the second, but using the default MAM3 aerosol module of CAM5.3 in CESM. The

purpose for designing the third group is to cross-validate the simulated DMS impacts on the ISE-induced CRE in the second
group. One bonus of the third group is to quantify the impacts of using different aerosol modules with different aerosol mixing
states on the simulated results. The anthropogenic emissions for MARC are slightly different from those for MAM3. All of the
experiments are summarized in Table 1.
**3 Results**
**3.1 DRE of ISE**

The all-sky DRE of various aerosol species from ISE is diagnosed as the difference between ShipRef_DMSRef and

ShipZero_DMSRef and shown in Figure 3. The total ISE can cause a global negative (cooling) DRE of −23.5 mW m−2, with the
strongest negative (cooling) DRE in the areas with intense shipping tracks, such as mid-latitude areas in the Pacific Ocean and
Atlantic Ocean, South China Sea, North Indian Ocean, and the Red Sea. The sulfate aerosols in the accumulation mode (i.e.,
ACC) contribute 89% to the global total DRE, followed by MOS aerosols with a contribution of 22%. Note that OC and MBS



has a counteracting warming effect (Remember that all gas-phase and aerosol emissions from shipping have been removed in
ShipZero scenarios). The contributions of other aerosol species are very limited and their magnitudes are smaller than 6%. The
magnitude of the total cooling effect is within the range from −50 to −10 mW m−2 estimated in previous studies (Endresen et al.,
2003; Schreier et al., 2007). The meridional variations of global zonally-averaged total DRE show that the DRE has the strongest
cooling effect of −80 mW m−2 between 30°N and 40°N and becomes weaker towards both polar regions and can be ignored
beyond 45°S and 60°N. The all-sky DRE of total aerosols in ShipLow_DMSRef and ShipHigh_DMSRef have similar patterns to
those in ShipRef_DMSRef and have magnitudes of +1.0 and −33.0 mW m−2, respectively. All of the calculated global DRE
values except for BC are confident at the 90% level.

**3.2 CRE of ISE under various shipping emission regulations**

The CRE of ISE is much stronger than the DRE and shows different spatial patterns under various shipping emission
regulations (Figure 4). At the reference level of shipping emissions (*ShipRef_DMSRef*), significant cooling CRE in SW is
simulated in areas of intense shipping tracks, such as the mid-latitude Pacific Ocean and the Baffin Bay between Canada and
Greenland, with a global average of −0.218 W m$^{-2}$. The LW CRE shows positive values in some small areas of high latitude,
with a global average of +0.065 W m$^{-2}$. Consequently, the global net CRE (SW+LW) is −0.153 W m$^{-2}$ with a similar spatial
pattern to that of SW. At the high level of shipping emissions (*ShipHigh_DMSRef*), the CRE changes to −0.253, +0.073, −0.179
W m$^{-2}$ for SW, LW, and net, respectively; more areas show significant changes than in *ShipRef_DMSRef*. Note that all of the
above values are statistically significant above the 90% confidence level. However, at the low level of shipping emissions
(*ShipLow_DMSRef*), fewer areas demonstrate significant changes than in *ShipRef_DMSRef* and *ShipHigh_DMSRef* and the
global averages of the CRE are not significant at the 90% confidence level for SW and net. These results indicate that more
stringent shipping emission regulation on sulfur content proposed by the IMO to be applied after 2020 could effectively reduce
or even largely eliminate the net CRE induced by ISE.
Further analyses demonstrate that the changes in CRE are caused by perturbations in both cloud water path (CWP;
Figure S1) and column-integrated cloud droplet number concentrations (CDNC; Figure 5) induced by ISE. Figure S1
demonstrates significant increases in total CWP mainly over the NPO and NAO at the reference and high levels of ISE. The
increases in total CWP is largely (87%) attributed to liquid CWP with the remaining contribution (13%) from ice CWP at the
reference shipping emission level. Such increases in CWP could reflect more solar radiation to space and thus cause a cooling
radiative effect at TOA, as shown in Figure 4. Note that very limited areas in North Pacific Ocean show significant increases in
ice CWP, indicating that a small portion of surface shipping emissions could be vertically transported to very high altitude and
form ice cloud. At the high level of shipping emissions, a larger increase in CWP is simulated, which is consistent with the
cooler radiative effects. However, no significant changes are simulated in total, ice, or liquid CWP at the low level of shipping
emissions. Associated with increases in CWP, the column-integrated cloud droplet number concentration (CNDC) also illustrates
significant increases at all levels of shipping emission (Figure 5), which collocate with increases in CWP (Figure S1) and
decrease in CRE (Figure 4) over the NPO and NAO. The sulfate aerosols from shipping emissions are highly efficient cloud
condensation nuclei (CCN) and thus can increase the CDNC. Note that cloud area fraction does not demonstrate any significant
changes due to shipping emissions (not shown).

**3.3 CRE of ISE under various DMS emissions**

The biogenic emissions of DMS over oceans can be oxidized to sulfates and compete against shipping emitted sulfates
for CCN and thus could influence the ISE-induced CRE. We find that the shipping emission-induced CRE exhibits significantly



different patterns and global averages at different emission levels of DMS (Figure 6). With DMS emissions ranging from the
reference level to low and zero levels, the magnitude of the ISE-induced negative CRE at SW increases from 0.218 to 0.457 and
2.435 W m$^{-2}$ on global scale, respectively; significant negative CRE is simulated over more areas in the SO. For CRE at LW,
more areas with significant warming are seen in the SO, NPO, and NAO, with the global averages change from +0.065 to +0.073
and +0.253 W m$^{-2}$ when DMS emissions changes from the reference to low and zero levels, respectively. For net CRE, it shares
the similar features with those at SW, but with smaller magnitudes.
The DMS emissions influence the ISE-induced CRE by perturbing the ISE-induced changes in CWP and column-
integrated CDNC (Figures S2 and 7). The shipping emission-induced changes in the total and liquid CWP increase as DMS
emission decreases, particularly over the SO, the NPO, and the NAO, while no significant changes are simulated in the ice CWP.
The increased CWP is closely associated with the increases in the column-integrated CDNC, which changes from 0.305×10$^9$ m$^{-2}$
to 0.476×10$^9$ m$^{-2}$ and 0.999×10$^9$ m$^{-2}$ on global scale as DMS emission decreases. The most prominent increases in CDNC are
seen in the SO, NPO, and NAO. These results suggest important roles of DMS emissions playing in modulating the ISE-induced
changes in cloud properties and radiation. Note that cloud area fraction does not illustrate any significant change (not shown).
As demonstrated in the above analysis, the impacts of DMS emissions on cloud response to shipping emissions are the
most prominent over the SO, the NPO, and the NAO, so further analysis are performed over these three regions. Figure 8 shows
cloud responses to shipping emission at different DMS emission levels over the three oceanic regions. Generally, cloud
responses become weaker and weaker as DMS emission increases from zero (*DMSZero*) to low (*DMSLow*) and reference
(*DMSRef*) level over all of the three regions. The most prominent change in cloud response is over the NPO, followed by the
NAO and the SO, which is probably due to the higher contribution of shipping emission to the total sulfur dioxide and sulfate
aerosols over the NPO than the NAO and the SO (Figure 2b and 2e). The removal of DMS emission (*DMSZero*) has a much
stronger influence on cloud response to shipping emission than reducing DMS emission by half (*DMSLow*), indicating a strong
non-linear competing effect for CCN between DMS and shipping emission.
**3.4 CRE of DMS under various shipping emissions**
Similar to DMS emissions' impacts on the ISE-induced CRE and cloud properties, ISE could also influence the DMS
emission-induced CRE and cloud properties. Generally, stronger cooling net CRE (−7.518 vs. −5.611 W m$^{-2}$) induced by DMS
emissions are seen when shipping emissions are ignored, particularly in areas of intense shipping tracks, such as over the NPO
and the NAO (Figure 9). Such a net cooling CRE is mainly the result of SW CRE. Stronger cooling CRE is associated with
larger increases in liquid and total CWP (Figure S3) and column-integrated CDNC (Figure S4) in simulations without shipping
emissions than those with shipping emissions.
It is worth pointing out that DMS emissions have significant warming CRE at LW, particularly over mid- and high-
latitude regions in the Southern Hemisphere and high-latitude regions in the Northern Hemisphere regardless of the presence of
shipping emissions (Figure 9). Such a warming CRE could be attributed to increases in total cloud area fraction, which is further
attributed to increases in the middle and low cloud area fraction in the high-latitude regions in both hemispheres (Figure S5). Our
results indicate that DMS is a significant source to CCN in the extremely clean polar regions in both hemispheres.
The area-averaged cloud responses over the SO, the NPO, and the NAO, to DMS emissions at different shipping
emission levels are shown in Figure 10. Cloud responses to DMS are stronger over the SO than over the NPO and the NAO
regardless of the presence of shipping emissions due to the fact that DMS and shipping emissions respectively dominate the
sulfur concentrations over the SO, and the NPO and the NAO (Figure 2). Moreover, cloud responses to DMS emissions become
much stronger over all of the three oceanic regions when shipping emissions have been removed. However, such changes in





cloud responses to DMS due to removal of shipping emissions (i.e., the slopes of the curves) are stronger over the NPO and the
NAO than over the SO, which is caused by very limited shipping emissions over the SO. These results again indicate a strong
non-linear competing effect for CCN between DMS and shipping emission.
**3.5 Impacts of choice of aerosol module on the results**

Besides the impacts of various ISE regulations and DMS emissions on the ISE-induced CRE, various assumptions

about the aerosol mixing states could also have an impact. Figure 11 shows the same results as Figure 6 but using MAM3 aerosol
module instead of MARC. At the reference level of DMS emissions (*DMSRef*), the ISE-induced CRE are generally stronger in
MAM3 (SW: −0.319, LW: +0.064, and Net: −0.255 W m$^{-2}$; Figure 11) than in MARC (SW: −0.218, LW: +0.065, net: −0.153 W
m$^{-2}$; Figure 6). More areas with significant cooling CRE are simulated in MAM3 than in MARC, particularly in the Atlantic
Ocean, West Pacific Ocean, and North Indian Ocean. At the low level of DMS emissions (*DMSLow*), both the global averages
and spatial patterns of the CRE in MAM3 are very similar to those in MARC. The two aerosol modules show the biggest
differences in CRE when DMS emissions are excluded (*DMSZero*). MARC simulates strong ISE-induced CRE over tropical
regions and the subtropical and mid-latitude areas of the SO, while MAM3 gives no significant CRE over these regions.
Generally, the ISE-induced CRE is stronger in MARC than in MAM3 when DMS emissions are excluded. The associated
changes in CDNC and CWP due to ISE illustrates similar patterns to changes in CRE (Figures S6 and S7). A possible reason for
such differences is the various mixing assumptions about sulfate and sea salt aerosols in MARC (external mixing) versus in
MAM3 (internal mixing) and warrants further studies.

By comparing Figure 8 with Figure 12 we also observe significantly different impacts of DMS emissions on cloud

response to shipping emissions (i.e., the slopes of these curves). MAM3 simulates a weaker impact of DMS emissions on cloud
response to shipping emissions than MARC, indicating a weaker non-linear competing effect for CCN between DMS and
shipping emissions in MAM3 than MARC.
**4 Conclusions and Discussion**

Aerosols from ISE could exert significant cooling on the Earth's climate system through aerosol–cloud and aerosol–

radiation interactions. To reduce the pollution and climatic effects from this emission source, the IMO set various emission caps
on sulfur content of marine fuel oil to be implemented in the future. Using a state-of-the-art climate model, we find that the
newly proposed more stringent emission regulations of shipping emissions can effectively reduce the ISE-induced CRE. As
demonstrated in our results, reducing sulfur contents from 3.5% to 2.7% and 0.5% could reduce both DRE (from −51.4 to −36.7
and −3.9 mW m$^{-2}$) and CRE (from −0.179 to −0.153 and −0.001 W m$^{-2}$) due to ISE, respectively. Although the ISE-induced
CRE would be insignificant on a global scale if sulfur contents of ship fuels were reduced to 0.5%, over some regions significant
CRE can still be detected—e.g., high latitude regions of the eastern Pacific Ocean. Therefore, implementation of cleaner fuels in
shipping sector, such as natural gas, could be a potential solution for completely eliminating sulfate-induced CRE.

More importantly, we find that the magnitude and regional spatial pattern of the ISE-induced CRE are highly sensitive

to natural DMS emissions. With DMS emissions reducing from 18.2 to 9.1 Tg S yr$^{-1}$ and zero, the ISE-induced net CRE changes
from −0.153 to −0.384 and −2.182 W m$^{-2}$, respectively. On the other hand, the DMS-induced net CRE changes from −5.611 to
−7.518 W m$^{-2}$ when shipping emissions at the reference level are removed in the simulations. It is worth noting that DMS is a
significant source to CCN in the extremely clean polar regions in both hemispheres. The strong interactions of CRE between
DMS and shipping emissions can be attributed to the nonlinearity in the responses of cloud processes to aerosols, particularly the




aerosol activation parameterizations (Abdul-Razzak et al., 1998). In a relatively clean environment, activated aerosol number
concentration increases as ambient aerosol number concentration increases until reaching a peak at a specific aerosol number
concentration, after which it decreases as ambient aerosol number concentration increases unless ambient vapor concentration is
drastically increased. In other words, the fraction of activated aerosols decreases as ambient aerosol concentration increases
(Figure S8). From the perspective of simulation, this nonlinearity in aerosol activation strongly suggests a reevaluation of CRE
induced by shipping and DMS emissions. From the perspective of field measurements of aerosol–cloud relationships, it warrants
careful attention when selecting measurement locations—shipping emissions-related measurements should be collected along
intense shipping tracks while in areas with as little DMS emissions as possible to avoid contamination from DMS, and vice
versa. Moreover, locations containing both shipping and DMS emissions should also be identified and sampled, in order to
investigate non-linear interactions between the emissions.
Finally, we find that two different aerosol schemes, with different representations of aerosol mixing state, could produce
a large difference (about 67%) in the ISE-induced global CRE. Generally, MARC aerosol module shows stronger nonlinear
cloud response to DMS and shipping emissions than MAM3. Overall, numerical studies on the uncertainties in the shipping
emission-induced CRE due to various ISE regulations, aerosol interactions, and aerosol mixing states can provide useful
information for policy makers and have implications for future projections of anthropogenic climate change.
Besides the above-mentioned contributors to the uncertainty in estimating the CRE induced by shipping emissions,
spatial resolution of the model is another significant source of this uncertainty. Possner et al. (2016) found that the ship-induced
shortwave CRE could increase by a factor of two as model spatial resolution decreases from 1 km to 50 km. With higher spatial
resolution, models can resolve fine-scale dynamical processes and feedbacks, such as interaction between aerosol and cumulus
clouds (Malavelle et al., 2017). Though model resolution-induced uncertainty is not the focus of this study, it should be taken
into account when interpreting the spread of shipping-induced CRE in studies of multi-model comparison.
Though we employed a state-of-the-art climate model in this study, it is not without caveats, given that neither MARC
nor MAM (including both MAM3 and MAM7 in CAM5) aerosol modules treat nitrate aerosols because of the high
computational expense for related aerosol-gaseous chemistry and aerosol thermodynamics calculations (Liu et al., 2012). Lack of
treating nitrate aerosols could result in uncertainties in our results based on the fact that both mass of nitrate aerosols emitted
from international shipping (e.g., Righi et al., 2011) and their hygroscopicity values (e.g., Kawecki and Steiner, 2018) are very
similar to those of sulfate aerosols, and thus nitrate aerosols could have non-negligible competing effects on CRE with sulfate
aerosols. Despite that some of the results from this study could be used to qualitatively project the potential outcome, a
quantitative assessment should be facilitated to address this topic with an improved model.
**Acknowledgments**
This study is supported by Concawe, whose mission is to conduct research on environmental issues relevant to the oil refining
industry. This research is also partially supported by the U.S. National Science Foundation (AGS-1339264), the U.S. Department
of Energy (DE-FG02-94ER61937), and the National Research Foundation (NRF) of Singapore under its Campus for Research
Excellence and Technological Enterprise programme. The Center for Environmental Sensing and Modeling (CENSAM) is an
interdisciplinary research group of the Singapore-MIT Alliance for Research and Technology (SMART). The CESM project is
supported by the National Science Foundation and the Office of Science (BER) of the U.S. Department of Energy. We would
like to acknowledge high-performance computing support from Yellowstone (ark:/85065/d7wd3xhc) provided by NCAR's
Computational and Information Systems Laboratory, sponsored by the National Science Foundation.




**Conflict of interest**
The authors declare no competing interests.
**Data and code availability**
The MARC source code is available via https://github.mit.edu/marc/marc_cesm/ and also archived with DOI
10.5281/zenodo.1117370, along with documentation on how to install and run the model. The commit 23e08fe was used in this
study. All the analysis code and model output data analyzed are available via
https://drive.google.com/drive/folders/1GHjrpvO06mzC8iFyT0Eif0PRiI56cReV?usp=sharing.

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




**Table 1. Summary of experiments.**

| Aerosol Modules | Experiments | DMS emissions | Ship emissions | Description |
|---|---|---|---|---|
| MARC | Shipping | *DMSRef* | *ShipZero* | DMS emission (Tg S yr$^{-1}$): |
| | | | *ShipLow* | DMSZero: 0 |
| | | | *ShipRef* | DMSLow: 9.1 |
| | | | *ShipHigh* | DMSRef: 18.2 |
| | DMS | *DMSZero* | *ShipZero* | |
| | | | *ShipRef* | |
| | | *DMSLow* | *ShipZero* | Ship emission (Tg S yr$^{-1}$): |
| | | | *ShipRef* | ShipZero: 0 |
| MAM3 | DMS | *DMSRef* | *ShipZero* | ShipLow: 1.0 (0.5%) |
| | | | *ShipRef* | ShipRef: 5.4 (2.7%) |
| | | *DMSZero* | *ShipZero* | ShipHigh: 7.0 (3.5%) |
| | | | *ShipRef* | |
| | | *DMSLow* | *ShipZero* | |
| | | | *ShipRef* | |

Notes: in *ShipZero* experiments, emission rates of all gas-phase and aerosol species from shipping emissions are set to zero;
while in *ShipLow*, *ShipRef*, and *ShipHigh* experiments, all shipping emission rates (such as OC and BC) are set to observations
except for emission rates of sulfur compounds (i.e. $SO_2$ and $SO_4$) which are modified.

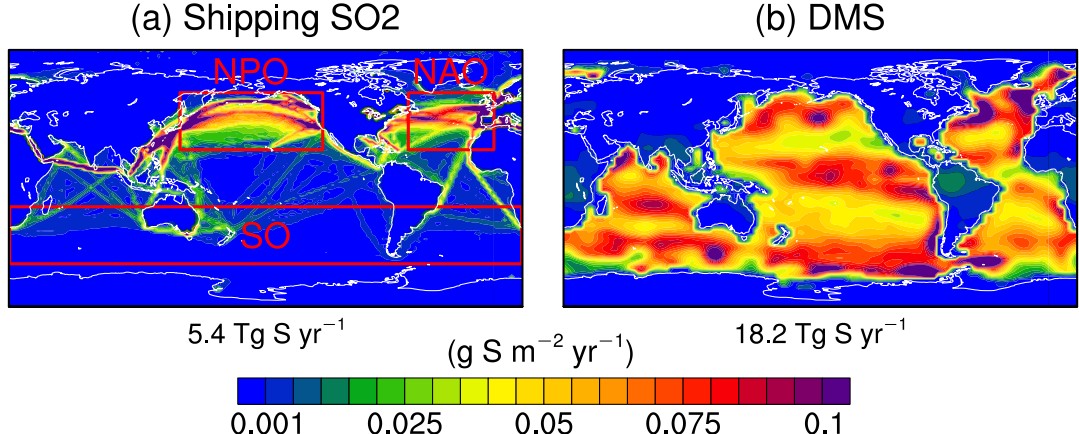

**Figure 1. Spatial patterns of annual means of sulfur emission (g S m$^{-2}$ yr$^{-1}$) from (a) international shipping and (b) natural DMS in the**
**simulation at the reference emission level (i.e., *ShipRef_DMSRef*). The numbers below each panel are the global total annual emissions.**
**Three regions are selected for further analysis: The North Pacific Ocean (NPO; 20°N–60°N, 140°E–240°E), the North Atlantic Ocean**
**(NAO; 20°N–60°N, 300°E–360°E), and the Southern Ocean (SO; 20°S–60°S, 0°E–360°E).**



**Figure 2.** Spatial patterns of (a) annual mean concentrations of total SO₂ (units: parts per billion by volume; ppbv), (b) and (c) are
respectively the contributions of shipping emission and natural DMS to total SO₂ in the lowest model layer. (d)–(f) are the same as (a)–
(c), but for sulfate aerosols. These results are from MARC simulations and calculated as the differences between the simulations with
the international shipping and DMS emissions at the reference and zero levels (i.e., *ShipRef_DMSRef* minus *ShipZero_DMSRef* and
*ShipRef_DMSRef* minus *ShipRef_DMSZero*).





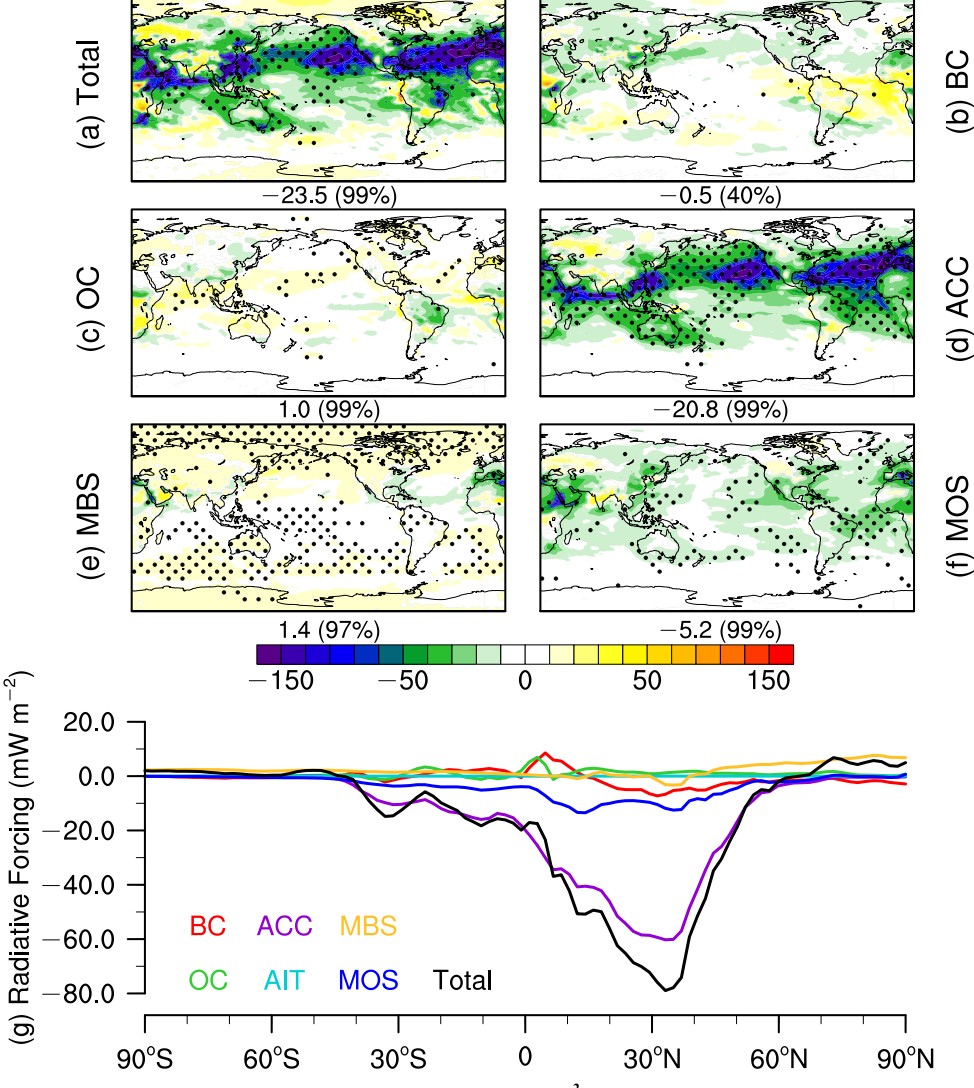


**Figure 3.** Simulated direct radiative effect (DRE; units: mW m$^{-2}$) of ISE at TOA by MARC. The DRE is calculated as the difference
between simulation results with and without ISE (i.e., *ShipRef_DMSRef* minus *ShipZero_DMSRef*) and averaged over the 30-year
period of simulations at all-sky conditions. Panels (a)–(f) show the spatial patterns of DRE due to ISE with the global mean differences
and the associated significant levels indicated by the numbers bellow each panel and panel (g) is the meridional variations of zonal
mean DRE for various aerosol types from ISE and their total effects. The expansions of the abbreviations can be found in Section 2.3.
The black dots represent grid points that are statistically significant above the 90% confidence level based on the two-tailed Student's
*t*-test.



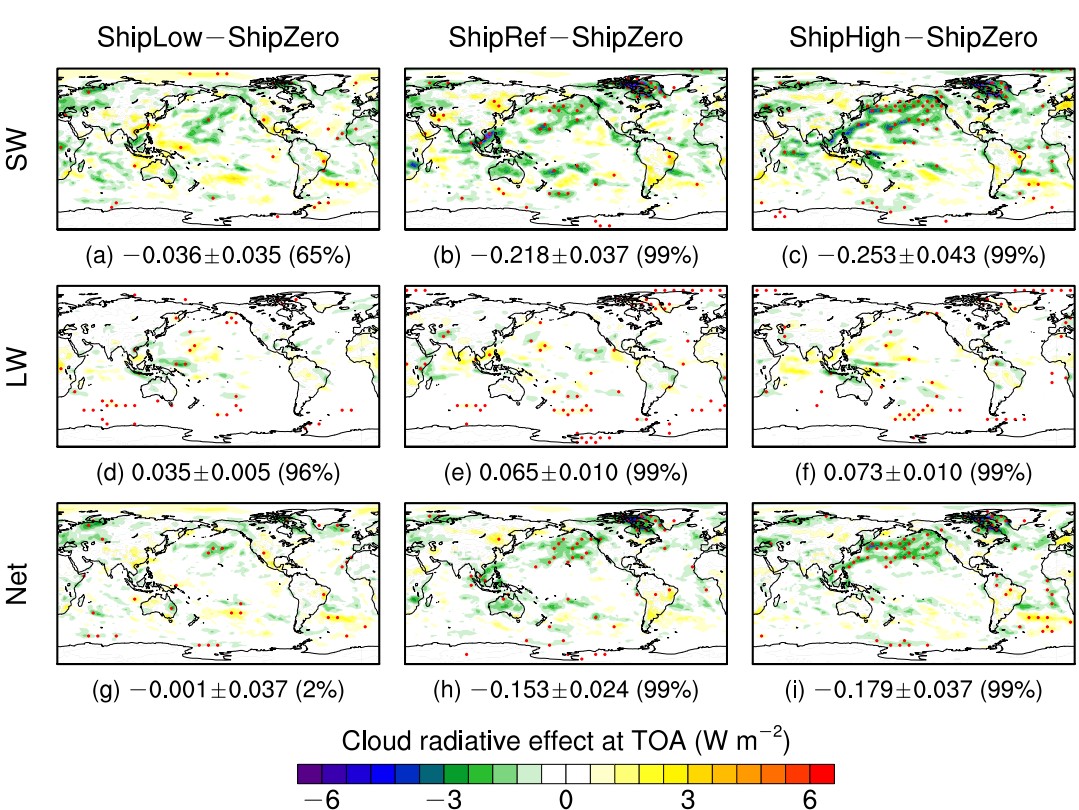

(a) −0.036±0.035 (65%)     (b) −0.218±0.037 (99%)     (c) −0.253±0.043 (99%)

(d) 0.035±0.005 (96%)     (e) 0.065±0.010 (99%)     (f) 0.073±0.010 (99%)

(g) −0.001±0.037 (2%)     (h) −0.153±0.024 (99%)     (i) −0.179±0.037 (99%)

**Figure 4.** Spatial patterns of MARC simulated cloud radiative effect (CRE; units: W m$^{-2}$) at TOA of ISE with various shipping emission levels. The CRE is calculated as the differences of radiation flux at TOA and at all-sky conditions between the simulation without shipping emissions and three simulations with the same DMS emissions at the reference level but various shipping emission levels (i.e., low, reference, high) in short-wave (SW), long-wave (LW), and net (SW+LW) and averaged over the 30-year simulation period. The numbers below each panel are the global means, standard deviation across the 30-year period, and the confidence level. The red dots represent grid points that are statistically significant above the 90% confidence level based on the two-tailed Student's *t*-test.




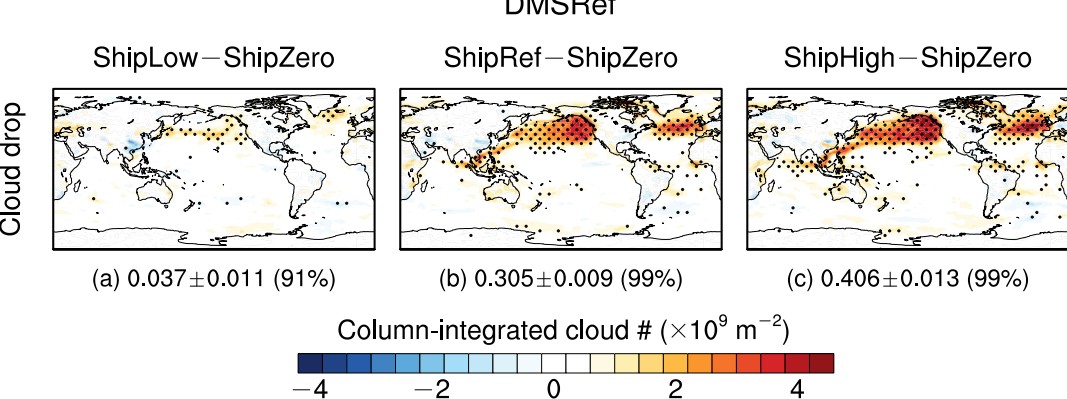

(a) 0.037±0.011 (91%)  (b) 0.305±0.009 (99%)  (c) 0.406±0.013 (99%)

**Figure 5.** Spatial patterns of MARC simulated column-integrated cloud droplet number concentration ($\times 10^9$ m$^{-2}$) response to international shipping emissions. The responses are calculated as the differences of cloud droplet number integrated through the whole atmospheric columns between the simulation without shipping emissions and three simulations with the reference shipping emission and various DMS emissions (i.e., zero, low, and reference) over the 30-year simulation period. The numbers below each panel are the global means, standard deviation across the 30-year period, and the confidence level. The black dots represent grid points that are statistically significant above the 90% confidence level based on the two-tailed Student's *t*-test.




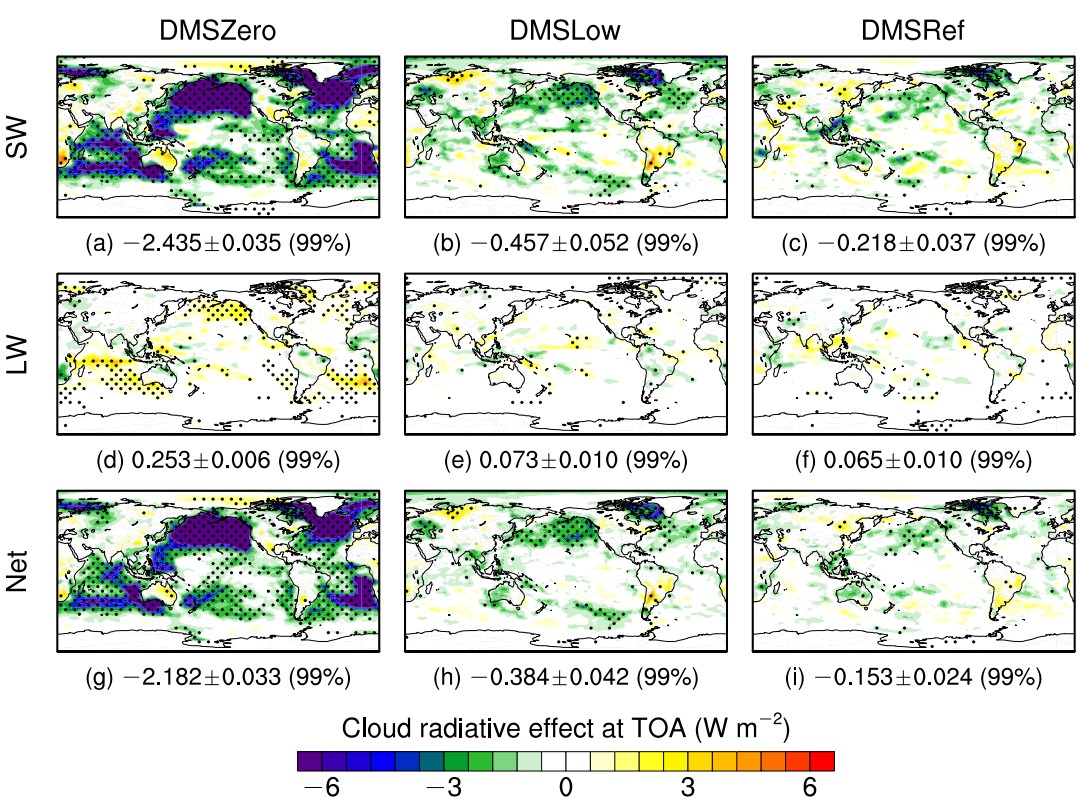

**Figure 6. Spatial patterns of MARC simulated cloud radiative effect (CRE; units: W m⁻²) at TOA of ISE at various DMS emission levels. The CRE is calculated as the differences of radiation flux at TOA and at all-sky conditions between the simulation without shipping emissions and three simulations with the same shipping emissions at the reference level but various DMS emission levels (i.e., zero, low, and reference) in short-wave (SW), long-wave (LW), and net (SW+LW) and averaged over the 30-year simulation period. The numbers below each panel are the global means, standard deviation across the 30-year period, and the confidence level. The black dots represent grid points that are statistically significant above the 90% confidence level based on the two-tailed Student's t-test.**





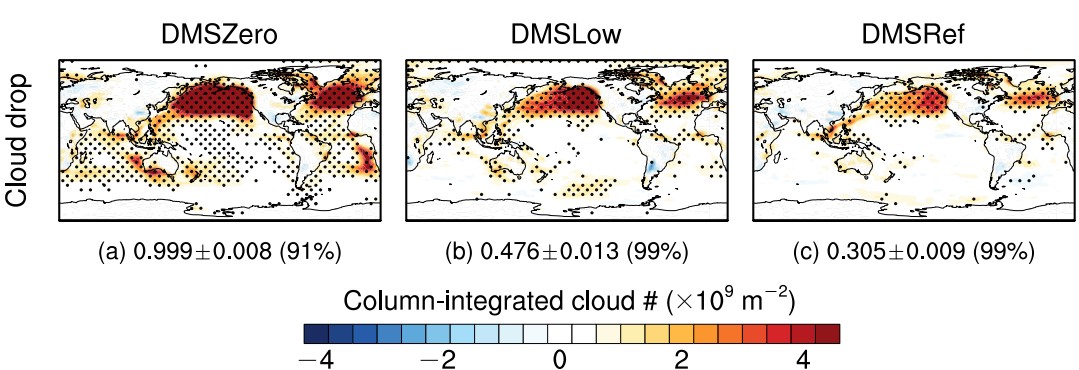

**Figure 7.** Spatial patterns of MARC simulated column-integrated cloud droplet number concentration ($\times 10^9$ m$^{-2}$) response to international shipping emissions. The responses are calculated as the differences of cloud droplet number integrated through the whole atmospheric columns between the simulation without shipping emissions and three simulations with the reference shipping emission and various DMS emissions (i.e., zero, low, and reference) over the 30-year simulation period. The numbers below each panel are the global means, standard deviation across the 30-year period, and the confidence level. The black dots represent grid points that are statistically significant above the 90% confidence level based on the two-tailed Student's $t$-test.

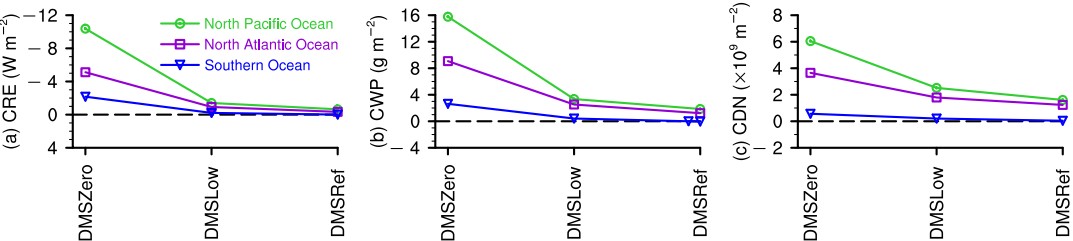

**Figure 8.** Impacts of DMS emissions on cloud responses to international shipping emissions. (a) Cloud radiative effects at TOA (W m$^{-2}$), (b) column-integrated cloud water path (g m$^{-2}$), and (c) column-integrated cloud droplet number ($\times 10^9$ m$^{-2}$). The green, purple, and blue curves respectively represent quantities area-averaged over the North Pacific Ocean (NPO), the North Atlantic Ocean (NAO), and the Southern Ocean (SO), which are shown as red boxes in Figure 1a. These results are from MARC simulations.



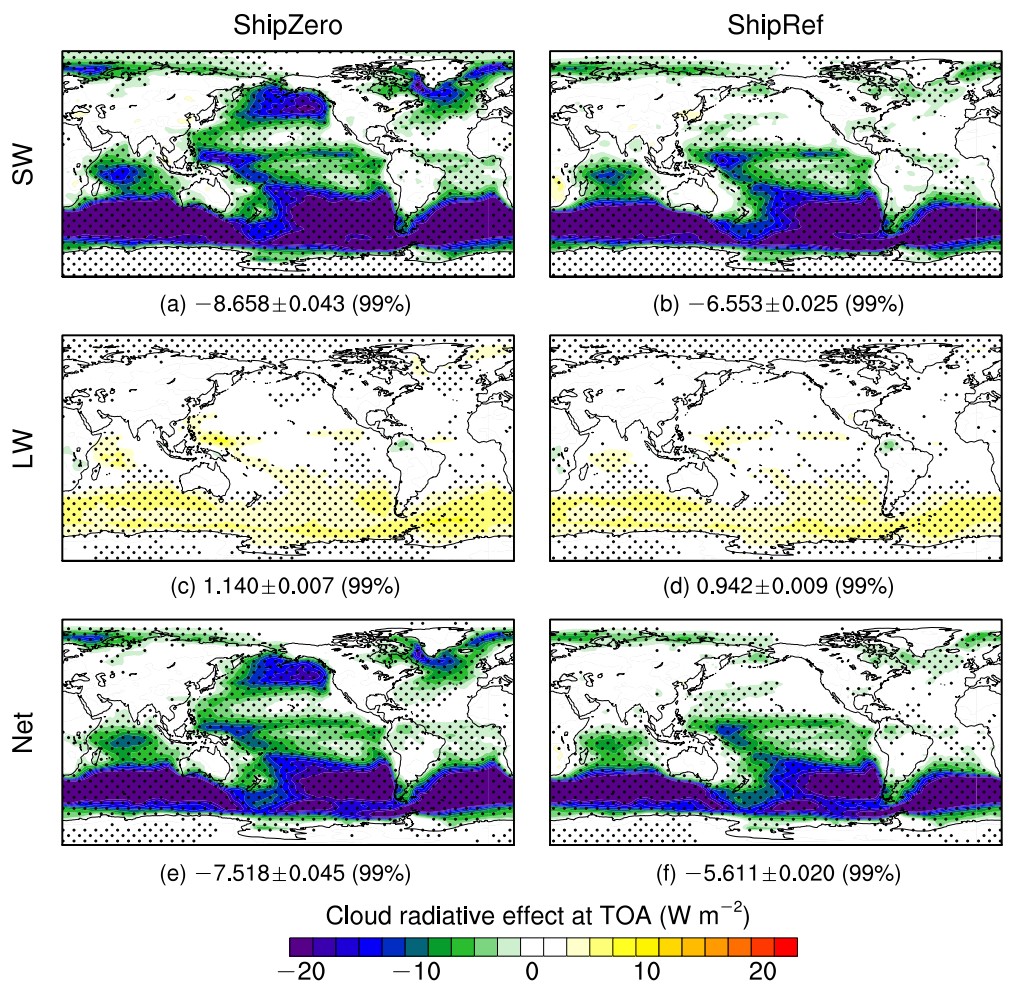

**Figure 9. Spatial patterns of MARC simulated cloud radiative effect (units: W m⁻²) of DMS emissions at various shipping emission levels. The CRE is calculated as the differences of radiation flux at TOA and at all-sky conditions between the simulation without DMS emissions and two simulations with the same DMS emissions at the reference level but various shipping emission levels (i.e., zero and reference) in short-wave (SW), long-wave (LW), and net (SW+LW) and averaged over the 30-year simulation period. The numbers below each panel are the global means, standard deviation across the 30-year period, and the confidence level. The black dots represent grid points that are statistically significant above the 90% confidence level based on the two-tailed Student's t-test.**

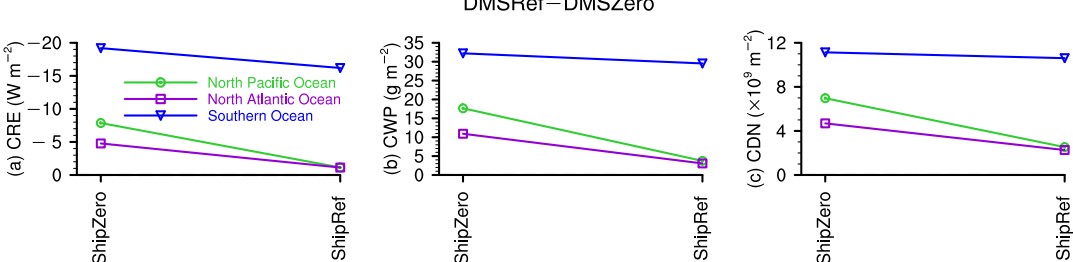

**Figure 10. Impacts of ISE on cloud responses to DMS emissions. (a) Cloud radiative effects at TOA (W m⁻²), (b) column-integrated cloud water path (g m⁻²), and (c) column-integrated cloud droplet number (×10⁹ m⁻²). The green, purple, and blue curves respectively represent quantities area-averaged over the North Pacific Ocean (NPO), the North Atlantic Ocean (NAO), and the Southern Ocean (SO), which are shown as red boxes in Figure 1a. These results are from MARC simulations.**





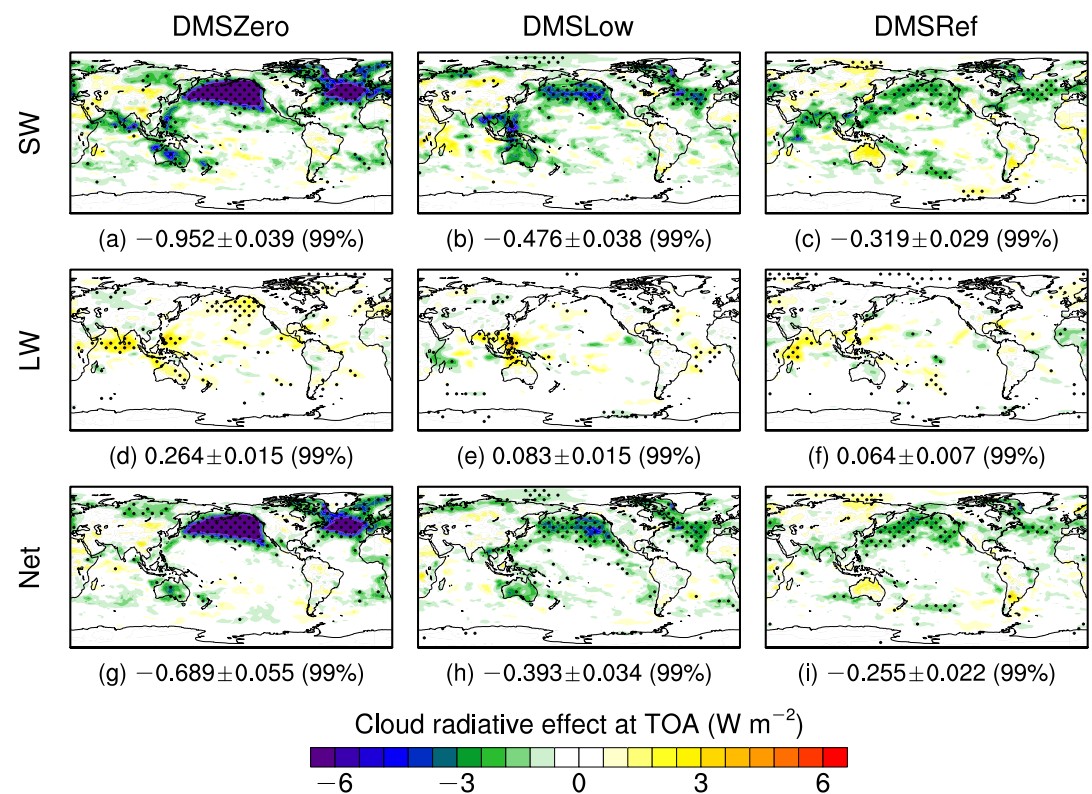

**Figure 11. Same as Figure 6, but using a different aerosol module, namely MAM3 rather than MARC.**

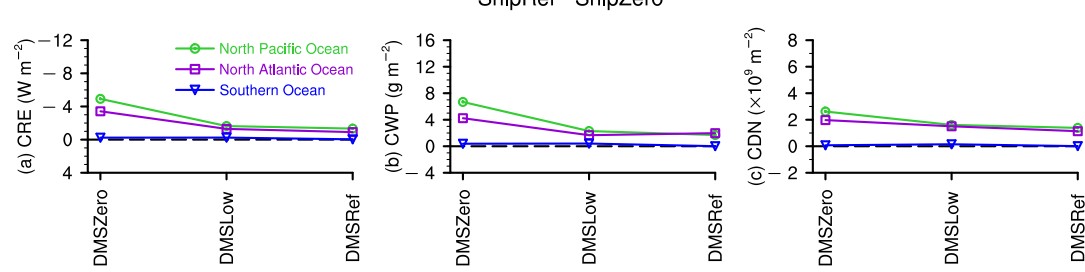

**Figure 12. Same as Figure 8, but using a different aerosol module, namely MAM3 rather than MARC.**