# Peer review of "Impacts on cloud radiative effects induced by coexisting aerosols converted from international shipping and maritime DMS emissions"

_Atmospheric Chemistry and Physics, 2018_

## Referee Comment (RC1) · Anonymous Referee #1 · 28 Jul 2018

Jin et al. used an Earth system model with two aerosol schemes that differ in size modes and mixing assumption to study the impact of international shipping emissions (ISE) and natural DMS emissions on cloud radiative effects (CRE) over vast oceanic regions. They found that the regular ISE emissions have a significant global net CRE, which can be further enhanced in a model configuration with reduced DMS emissions. The study also demonstrated that the different aerosol treatments can influence the magnitude and spatial pattern of ISE-induced CRE. The authors suggest a re-evaluation of the ISE-induced CRE with the DMS variability considered. The impact of ISE on CRE is very uncertain. The findings of this study partially explain why the magnitude of ISE-induced CRE has a large spread, shown in the literature. The

paper is well written in general and results are clearly presented. However, there are some places in the manuscript that would benefit from further clarification and improvements. I recommend it for publication after the following comments and suggestions are considered.

1) L31-37: Only sulfur emissions are mentioned in the literature review. How about primary particles such as black carbon (BC) and organic carbon (OC)? If both BC and OC from shipping are fixed at the standard emission rate in the various model experiments, please comment on the role of these primary particles, compared to the secondary sulfate converted from sulfur dioxide.

2) L74: I don't think the word "diagnose" is properly used here.

3) L78-79: This is inaccurate. Aerosol in CAM5 does not have a direct microphysical influence on convective clouds, but can have an impact indirectly.

4) L80-96: Are there other differences in aerosol-related treatments between MAM3 and MARC, for example, gas condensation, new particle formation, cloud processing including aqueous-phase chemistry and particle resuspension? What differentiate the BC, OC and sulfate mass for each of the relevant size modes upon emissions? Answers to these questions are critical to understanding the model results in this study (Sect. 3.5), so I suggest including them here.

5) L115-116: What is the purpose to treat BC and OC differently in ShipZero and the other three experiments? This is not clearly noted when interpreting the model results (e.g., Figs 4 and 5).

6) L148-150: Are the contributions by aerosol modes or types derived from radiation diagnostics? It is unclear to me whether the radiation diagnostics are done in such a detailed way (by aerosol types). It is counterintuitive to see positive DRE for OC but negative DRE for BC (Figure 3). Any explanations?

7) L196-197: Needs clarification on the three numbers. How do they compare to the

base case (e.g., Shipzero_DMSref)? It would be interesting to have some discussion about the relative changes in CDNC, compared to the role of sea salt and other types of aerosols.

8) L200-208: I wonder if clouds in any of these regions are more susceptible to DMS emissions than the others (i.e., relative forcing changes normalized by relative emissions changes).

9) L220: this sounds like an important claim. The role of any specific type of aerosol in affecting high-latitude clouds depends much on the background total aerosol concentrations. What's the model performance in simulating high-latitude natural and anthropogenic aerosols?

L230-242: The two aerosol schemes gave very different results on the magnitude of the ISE-induced CRE, which is my biggest concern. The current explanation is too vague. More in-depth analysis is required here. Have the two schemes been systematically compared in terms of the global aerosol direct and indirect forcing?

---

## Referee Comment (RC2) · Anonymous Referee #2 · 28 Jul 2018

This paper addresses the well-known and important topic of negative radiative forcing induced by aerosols formed from international shipping emissions. The paper is novel in addressing natural (DMS) emissions and shipping emissions of aerosols and aerosol precursors simultaneously to study their non-additive contributions to cloud formation and cloud radiative effects (CRE). As an interesting add-on, uncertainties due to microphysics modelling are investigated through the use of different aerosol modules and assumptions on mixing states. The paper is well written and the experiments well designed, properly reflecting the current state of science. I don't have any technical comments beyond those already pointed out by Referee #1

I recommend publication after the following issues have been addressed:

It can be misleading to present the cooling from ISE as balancing GHG warming. E.g. from reading the abstract some may be tempted to conclude that the IMO global sulfur cap from 2020 may contribute to global warming (through reduced CRE), and thus be the wrong way to go. Although global average net radiative forcing may indeed become more positive through this regulation, it should be made clear, at least in the conclusions, that the global sulphur cap is highly beneficial for air quality. The paper already contains relevant references for this (e.g. Corbett et al. 2007, Winebrake et al. 2009). Compensating for GHG warming through aerosol cooling is also problematic because the radiative forcing by aerosols is highly spatially variable. Interestingly in this regard, shipping-induced CRE seems to cause up to 3 W/m2 warming (Figure 4) over Central Europe and areas in China and South America. Finally the cooling contribution, as pointed out by this study, has a large uncertainty (while GHG warming is easier to estimate), and the impact on climate parameters (local temperature, precipitation, etc.) from CRE is even more uncertain than the impact on radiative forcing itself.

Section 2.4 requires some more text as it is important for correct interpretation of the results. For example, what do you mean by diagnostic and prognostic calls in this context? This may well be obvious to insiders of radiation modelling, but what does it imply for the results presented in this paper? Can effects calculated either in the diagnostic or prognostic calls be compared to each other? Also "In this way, the DRE and CRE of ISE can be isolated and evaluated separately" I don't quite understand this sentence.

line 50: is from –> ranges from

line 123: Why referring to Corbett et al., 2007 here? The 0.5% cap wasn't mentioned, and plans for 2020 were not known in 2007.

line 123: although "International Maritime Organization, 2016" looks like a good reference, in the list of references we only learn "IMO sets 2020 date for ships to comply

with low sulfur fuel oil requirement, 2016" which looks like a log rather than a reference. Is there a link to an accessible report or news release instead?

line 168: proposed by IMO –> decided by IMO

line 182: demonstrate –> exhibit or show

line 199: illustrate –> exhibit or show

line 201: analysis –> analyses

---

## Short Comment (SC1) · 4 Aug 2018

In their study, the authors investigate the uncertainties associated with estimating aerosol indirect effects (AIES) induced by global shipping emissions in a general circulation model. The uncertainties studied here are three-fold, in the sense that 1) AIEs from shipping emissions are shown to non-linearly depend on the background concentration of natural DMS emissions (an expected result and very valuable as it has not been quantified before), 2) AIEs from shipping emissions depend on the amount of sulphur contained in the fuel (as has been shown in earlier studies) and 3) estimated AIEs from shipping emissions depend heavily on the aerosol microphysics module applied

in the general circulation model.

As I have also worked on this topic in the past, I find the study extremely interesting and relevant and I have some comments/remarks concerning the results, the presented analysis and framing of the results presented here in the view of earlier studies.

Main point:

In my view, the discussion of the differences between the two aerosol modules and their impact on the results warrants a bit more investigation/explanation. In Peters et al (2012), P12 in the following, we investigated the uncertainties of AIEs from shipping emissions related to the assumed emission particle size distribution and the total amount of fuel burnt. We found a significant impact of the assumed particle size distribution on the estimated AIEs: assuming all sulphuric compounds being assigned to the soluble Aitken mode at point of emission (as supported by various field observations) leads to significantly more negative AIEs than assigning them half and half to the soluble Accumulation and Coarse modes as was done in the standard Aerocom emission setup (cf P12). This is due to the substantially higher number of primary emitted soluble particles. This was further substantiated in the corrigendum to P12 (Peters et al., 2013), in which a bugfix in the aerosol module lead to an even higher number of emitted particles. My question here is: what size mode are the shipping emissions (and the DMS emissions) assigned to at the point of emission in MAM3 and MARC? This is not described in Section 2, but is a critical point. Compare to the detailed analysis from emission all the way to the resulting effects on cloud properties detailed in Peters et al. (2012, 2014), because in the end, AIEs are the end product of a long chain of processes calculated in various parameterizations with, most certainly, inherent uncertainties. Are there diagnostics of aerosol numbers per mode (see P12) available for the current study so as to investigate the differences between then two aerosol microphysics modules in more detail? This would also help in investigating the interplay with different assumed DMS emission levels, especially for the case of zero DMS emissions where the differences between the two parameterisations are largest.

Can CCN diagnostics (see P12, Peters et al. 2014, P14 in the following) be provided to further investigate these points?

Minor points:

Lines 40-42: please also mention the Corrigendum to P12 – i.e. Peters et al 2013 - as the results from P12 suffered from a bug in the aerosol microphysics module. The results presented in Peters et al (2013) are thus more sound.

Lines 118 – 126: while investigating the effect of total sulfur content in bunker fuel is an important issue, please also mention uncertainties related to the total amount of fuel burnt (cf P12).

Lines 170-172: see my above comment regarding an analysis of the causal chain from emissions -> AIEs.

Lines 178-180: this reads like the increase in CWP leads to an increase in CDNC, but it should be the other way around (at least from the viewpoint of a parameterisation in which a causal connection of cause-and-effect has to be established by design)

Lines 181-182: this is obvious and is out of place at the end of this paragraph (compare to lines 30-34 in the Introduction)

Lines 187-192: This is a very interesting paragraph, specifically because DMS emissions are natural and thus an integral part of the climate system and are most probably also included when tuning the TOA radiation balance of the model. Capturing the "correct" background (pre-industrial) aerosol distribution is an extremely difficult task, see e.g. Stevens et al 2017 for the case of developing an aerosol climatology, and is critical for estimating anthropogenic AIEs (as is very nicely shown in this paper). Coming to the point, leaving out DMS emissions results in a quite large TOA radiative imbalance in excess of -5 Wm-2 (Figure 9). Although the model is constrained by prescribed SSTs, this imbalance, which is much larger on local scales, could have an effect on the results.

Lines 266-268: a very important point. Even more importantly, this calls for a re-evaluation of aerosol and cloud microphysics parameterisations in general circulation models.

Lines 269-272: I completely agree. In Peters et al 2011, P11, we applied a specific sampling routine to observational data in order to sample for the effect of shipping emission on cloud properties in "pristine" oceanic areas, where "pristine" was mean with regards to anthropogenic emissions. Looking at the maps displayed in Figure 2, shipping emissions are trumped by DMS emissions in two of the regions sampled in P11: the SE Pacific and the mid-Indian Ocean region. However, the third region investigated in P11, the mid Atlantic, shows a significant contribution of shipping emissions compared to DMS. We also focused more on that region in P14 and concluded that for "observational studies of AIEs, this highlights the ever so important and often discussed aspect of correctly defining the background ('pre-industrial') reference state against which to gauge the present-day observations." The present study thus very convincingly corroborates our conclusions drawn in 2014.

References:

Peters, K., J. Quaas, and H. Graßl (2011), A search for large‐scale effects of ship emissions on clouds and radiation in satellite data, J. Geophys. Res., 116, D24205, doi: 10.1029/2011JD016531.

Peters, K., Stier, P., Quaas, J., and Graßl, H.: Aerosol indirect effects from shipping emissions: sensitivity studies with the global aerosol-climate model ECHAM-HAM, At-mos. Chem. Phys., 12, 5985-6007, https://doi.org/10.5194/acp-12-5985-2012, 2012

Peters, K., Stier, P., Quaas, J., and Graßl, H.: Corrigendum to "Aerosol indirect effects from shipping emissions: sensitivity studies with the global aerosol-climate model ECHAM-HAM" published in Atmos. Chem. Phys., 12, 5985–6007, 2012, Atmos. Chem. Phys., 13, 6429-6430, https://doi.org/10.5194/acp-13-6429-2013, 2013.

Peters, K., Quaas, J., Stier, P. and Graßl, H. :Processes limiting the emergence of detectable aerosol indirect effects on tropical warm clouds in global aerosol-climate model and satellite data, Tellus B: Chemical and Physical Meteorology, 66:1, DOI: 10.3402/tellusb.v66.24054, 2014.

Stevens, B., Fiedler, S., Kinne, S., Peters, K., Rast, S., Müsse, J., Smith, S. J., and Mauritsen, T.: MACv2-SP: a parameterization of anthropogenic aerosol optical properties and an associated Twomey effect for use in CMIP6, Geosci. Model Dev., 10, 433-452, https://doi.org/10.5194/gmd-10-433-2017, 2017.

---

## Author Comment (AC1) · 5 Sep 2018

**Dear Editor,**

**Thank you for spending time in handling our manuscript. We also very much thank all of the reviewers for their thoughtful and constructive suggestions and comments that have helped us to improve the quality of the manuscript. Below are our point-by-point responses to each comment, which is in light-blue font color.**

**Anonymous Referee #1**

Jin et al. used an Earth system model with two aerosol schemes that differ in size modes and mixing assumption to study the impact of international shipping emissions (ISE) and natural DMS emissions on cloud radiative effects (CRE) over vast oceanic regions. They found that the regular ISE emissions have a significant global net CRE, which can be further enhanced in a model configuration with reduced DMS emissions. The study also demonstrated that the different aerosol treatments can influence the magnitude and spatial pattern of ISE-induced CRE. The authors suggest a re-evaluation of the ISE-induced CRE with the DMS variability considered. The impact of ISE on CRE is very uncertain. The findings of this study partially explain why the magnitude of ISE-induced CRE has a large spread, shown in the literature. The paper is well written in general and results are clearly presented. However, there are some places in the manuscript that would benefit from further clarification and improvements. I recommend it for publication after the following comments and suggestions are considered.

We appreciate the reviewer's recognition of the potential importance of our manuscript. We carefully revised our manuscript based on the reviewer's comments. The following are our point-by-point responses to these comments.

1) L31-37: Only sulfur emissions are mentioned in the literature review. How about primary particles such as black carbon (BC) and organic carbon (OC)? If both BC and OC from shipping are fixed at the standard emission rate in the various model experiments, please comment on the role of these primary particles, compared to the secondary sulfate converted from sulfur dioxide.

Thanks for raising this important point that we have missed. Yes, both BC and OC from shipping are fixed at the standard emission level in the simulations with shipping emissions turned on. Now we added the following sentence at the end of the second paragraph in the Introduction to address this concern:

"Note that although ISE also contain significant amount of black carbon and organic carbon aerosols, since this study mainly focuses on aerosol induced CRE instead of aerosol direct radiative effect, only primary and secondary sulfate as well as internal mixtures of sulfate and carbonaceous aerosols are addressed due to their much higher hygroscopicity than those of external black carbon and organic carbon aerosols (Pringle et al., 2010),"

2) L74: I don't think the word "diagnose" is properly used here.

We changed "diagnose" to "evaluate".

3) L78-79: This is inaccurate. Aerosol in CAM5 does not have a direct microphysical influence on convective clouds, but can have an impact indirectly.

We agree with the reviewer and now revised the sentence, as shown here:
"Similar to other climate models, CAM5 does not directly include aerosol's influence through microphysics on convective clouds, but it allows aerosols to influence convective clouds indirectly, such as by aerosol's direct effect on circulation, surface evapotranspiration and so on."

4) L80-96: Are there other differences in aerosol-related treatments between MAM3 and MARC, for example, gas condensation, new particle formation, cloud processing including aqueous-phase chemistry and particle resuspension? What differentiate the BC, OC and sulfate mass for each of the relevant size modes upon emissions? Answers to these questions are critical to understanding the model results in this study (Sect. 3.5), so I suggest including them here.

To address the reviewer's questions, we added the following paragraph as a new subsection of "2.4 Difference between MARC and MAM3".

"The most fundamental difference between MARC and MAM is that MARC includes both external and internal mixtures of aerosols in fifteen modes while MAM treats all aerosols as internal mixtures in three modes. As a result, the processes of many aerosol microphysical processes including gaseous condensation, new particle formation, and nucleation scavenging differ between these two models (Kim et al., 2008; Grandey et al., 2018; Rothenberg et al., 2018). "For instance, aerosol activation or nucleation scavenging in MARC and MAM3 is calculated based on competition for water vapor among various types (or modes) of aerosols with different hygroscopicity. In this case, external sulfate modes and the mixture of BC and sulfate (MBS) with BC as core and sulfate as shell would have the same hygroscopicity as sulfate, while external BC and OC would have much lower hygroscopic values. Whereas, MAM calculates this process based on the volume weighted hygroscopicity of each mode based on all the aerosol constitutions within the mode. In that case, the change of individual aerosol species would not influence much the number of activated aerosol substantially" (Rothenberg et al., 2018)."

We have added adequate descriptions of aforementioned differences in the manuscript, or in other cases, direct the reader to corresponding references.

5) L115-116: What is the purpose to treat BC and OC differently in ShipZero and the other three experiments? This is not clearly noted when interpreting the model results (e.g., Figs 4 and 5).

The purpose is to extract the total effects induced by the ISE, i.e., including BC, OC, and sulfur. Figures 4 and 5 are used to address the impacts of various sulfur caps of ISE on CRE. The ISE-induced CRE from various aerosol types is not separated. Only the radiative effects of various aerosol types are diagnosed and presented in Figure 3, such as for BC and OC.

6) L148-150: Are the contributions by aerosol modes or types derived from radiation diagnostics? It is unclear to me whether the radiation diagnostics are done in such a detailed way (by aerosol types). It is counterintuitive to see positive DRE for OC but negative DRE for BC (Figure 3). Any explanations?

Yes, the DRE of each aerosol mode is derived from radiation diagnostics. We double-checked the diagnosed DRE of BC at TOA in all simulations, which are positive at each grid of the globe. Therefore, the negative values of BC DRE in Figure 3 stems from the subtraction (i.e., *ShipRef_DMSRef* minus *ShipZero_DMSRef*). Adding shipping emissions could induce very slight change in the meteorological fields, such as winds, precipitation and so on, which may result in perturbations in the deposition of BC (including both anthropogenic BC over land and shipping BC) and consequently results in negative BC DRE of shipping emissions. It could be the same reason for positive OC DRE induced by shipping emissions.

7) L196-197: Needs clarification on the three numbers. How do they compare to the base case (e.g., Shipzero_DMSref)? It would be interesting to have some discussion about the relative changes in CDNC, compared to the role of sea salt and other types of aerosols.

We calculated the relative changes of column-integrated CDNC induced by shipping emissions by comparing to that in the base case, i.e., ShipZero_DMSRef. We revised the corresponding sentence to:

"The increased CWP is closely associated with the increases in the column-integrated CDNC, which changes from $0.305 \times 10^9$ m$^{-2}$ (2.5%, relative to climatological CDNC in *ShipZero_DMSRef* simulation) to $0.476 \times 10^9$ m$^{-2}$ (3.9%) and $0.999 \times 10^9$ m$^{-2}$ (8.3%) on global scale as DMS emission decreases. These results imply that sea salt and DMS emissions are the dominant sources of cloud seeds over remote oceans."

The contributions to total CDNC from shipping, DMS, sea salt, and transported aerosols from land are an interesting topic and should be qualified in future studies.

8) L200-208: I wonder if clouds in any of these regions are more susceptible to DMS emissions than the others (i.e., relative forcing changes normalized by relative emissions changes).

We appreciate the point from this reviewer, however, we have not collected data with finer temporal resolution in order to carefully calculate this parameter. We have only monthly model output, but it is a good point to check in the future.

9) L220: this sounds like an important claim. The role of any specific type of aerosol in affecting high-latitude clouds depends much on the background total aerosol concentrations. What's the model performance in simulating high-latitude natural and anthropogenic aerosols?

It is hard to evaluate the model performance in simulating high-latitude aerosols, because of a lack of observations (e.g., satellite do not retrieve AOD over very high-latitude regions). By simply comparing high-latitude aerosol loadings in MAM and MARC in another paper of us ( https://doi.org/10.5194/acp-2018-118), we found that generally, both models simulate very similar magnitude of total AOD in high-latitude; MARC has a slightly lower sea salt and sulfate loadings, a lightly higher BC loading than MAM in high-latitude.

10) L230-242: The two aerosol schemes gave very different results on the magnitude of the ISE-induced CRE, which is my biggest concern. The current explanation is too vague. More in-depth

analysis is required here. Have the two schemes been systematically compared in terms of the global aerosol direct and indirect forcing?

The two aerosols schemes have been systematically compared in another paper (https://doi.org/10.5194/acp-2018-118), including aerosol loadings for different aerosol modes, total aerosol optical depth, DRE, CCN, CDNC, and CRE. Differences in some of these variables are obvious. The similar analysis will be performed for shipping emissions. However, current model configuration in this manuscript and the above paper cannot achieve this goal, because some variables such as CCN are not in the model history files in the simulations designed for this manuscript.

To address this concern, we added the following discussion at the end of Section 3.5:
"To track down all the possible reasons for the differences in the ISE-induced CRE between the two aerosol schemes, more detailed analyses on a long chain of processes related to both aerosols and clouds are required, as done by Peters et al. (2014), which is out of the scope of this study and warrants more studies in the future."

**Anonymous Referee #2**

This paper addresses the well-known and important topic of negative radiative forcing induced by aerosols formed from international shipping emissions. The paper is novel in addressing natural (DMS) emissions and shipping emissions of aerosols and aerosol precursors simultaneously to study their non-additive contributions to cloud formation and cloud radiative effects (CRE). As an interesting add-on, uncertainties due to microphysics modelling are investigated through the use of different aerosol modules and assumptions on mixing states. The paper is well written and the experiments well designed, properly reflecting the current state of science. I don't have any technical comments beyond those already pointed out by Referee #1.

I recommend publication after the following issues have been addressed:

We thank the reviewer for his/her positive evaluation on our manuscript. We carefully revised the manuscript based on all comments and our point-by-point responses are listed below.

It can be misleading to present the cooling from ISE as balancing GHG warming. E.g. from reading the abstract some may be tempted to conclude that the IMO global sulfur cap from 2020 may contribute to global warming (through reduced CRE), and thus be the wrong way to go. Although global average net radiative forcing may indeed become more positive through this regulation, it should be made clear, at least in the conclusions, that the global sulphur cap is highly beneficial for air quality. The paper already contains relevant references for this (e.g. Corbett et al. 2007, Winebrake et al. 2009). Compensating for GHG warming through aerosol cooling is also problematic because the radiative forcing by aerosols is highly spatially variable. Interestingly in this regard, shipping-induced CRE seems to cause up to 3 W/m2 warming (Figure 4) over Central Europe and areas in China and South America. Finally the cooling contribution, as pointed out by this study, has a large uncertainty (while GHG warming is easier to estimate), and the impact on climate parameters (local temperature, precipitation, etc.) from CRE is even more uncertain than the impact on radiative forcing itself.

We appreciate the point of this reviewer and we can understand the concern. In the manuscript, we mainly focus on the ISE-induced negative CRE at TOA averaged over the globe and actually avoid concluding or implying that ISE could induce surface cooling specifically as a balance to GHG warming. On the other hand, we do not believe that it is improper for the reader to conclude from our results that the new IMO sulfur cap from 2020 may partially balance global warming. Additionally, in many places of this manuscript, such as in the first paragraph of Section 4, we did state that reducing the ISE can improve air quality. We also mentioned shipping-induced warming CRE over some regions, such as China and Central Europe. Overall, we believe that we are objective in interpreting our results.

Section 2.4 requires some more text as it is important for correct interpretation of the results. For example, what do you mean by diagnostic and prognostic calls in this context? This may well be obvious to insiders of radiation modelling, but what does it imply for the results presented in this paper? Can effects calculated either in the diagnostic or prognostic calls be compared to each other? Also "In this way, the DRE and CRE of ISE can be isolated and evaluated separately" I don't quite understand this sentence.

The reviewer's point is well taken. We have revised this paragraph by adding more detailed descriptions of calculation steps. Here, by diagnostic we mean the results from radiation calls are not propagated to any actual model physical and dynamical calculations rather than being recorded in output and, therefore, do not influence model integration in the next time step; while by prognostic we mean the results from radiation calls are not only recorded in model history output but also passed to following model calculations and, therefore, affect the results of actual integration.

In our model configuration, the calculations of DRE and CRE of ISE in the diagnostic mode are the same as those in prognostic mode, but diagnostic mode can output the DRE of ISE for each individual aerosol type, such as BC, OC, sulfates, and so on, which are shown in Figure 3.

"In this way, the DRE and CRE of ISE can be isolated and evaluated separately." By this sentence we mean "This is the way we isolate the DRE and CRE of ISE".

The revised paragraph is shown here:

"In the diagnostic mode of CESM-MARC, the DRE are diagnosed by calling the radiation scheme three times in each radiation time step. The first call does not include any aerosols, providing "clean-sky" diagnostics (Ghan, 2013). The second call includes only mineral dust and sea salt aerosols. The third call includes all aerosols. The first and third call are diagnostic, i.e. the radiation budget calculated from these two calls are only used to as model output, therefore they do not influence model integration in the next time step; while the second call is prognostic, i.e., the radiation budget from this call is passed to other model schemes to calculate associated model variables, such as temperature, surface evaporation and so on. Therefore, DRE of only dust and sea salt aerosols are prognostic while all other aerosols including ISE are diagnostic. Note that all radiation variables calculated in these three calls are stored in the model history files for further analyses. In the complimentary MAM3 simulations, the first radiation call (prognostic) includes all aerosols while the second call is a "clean-sky" diagnostic call, excluding all aerosols."

line 50: is from –> ranges from

Done.

line 123: Why referring to Corbett et al., 2007 here? The 0.5% cap wasn't mentioned, and plans for 2020 were not known in 2007.

Thanks for pointing this out. Now we removed this citation in the sentence.

line 123: although "International Maritime Organization, 2016" looks like a good reference, in the list of references we only learn "IMO sets 2020 date for ships to with low sulfur fuel oil requirement, 2016" which looks like a log rather than a reference. Is there a link to an accessible report or news release instead?

We apologize for missing the link to this media news. The link to a detailed report on low sulfur fuel oil requirement can be found here: http://www.imo.org/en/mediacentre/pressbriefings/pages/mepc-70-2020sulphur.aspx. We have also added this link to the references.

line 168: proposed by IMO –> decided by IMO

Done.

line 182: demonstrate –> exhibit or show

Done.

line 199: illustrate –> exhibit or show

Done.

line 201: analysis –> analyses

Done.

**Short Comment #1**

In their study, the authors investigate the uncertainties associated with estimating aerosol indirect effects (AIES) induced by global shipping emissions in a general circulation model. The uncertainties studied here are three-fold, in the sense that 1) AIEs from shipping emissions are shown to non-linearly depend on the background concentration of natural DMS emissions (an expected result and very valuable as it has not been quantified before), 2) AIEs from shipping emissions depend on the amount of sulphur contained in the fuel (as has been shown in earlier studies) and 3) estimated AIEs from shipping emissions depend heavily on the aerosol microphysics module in the general circulation model.

As I have also worked on this topic in the past, I find the study extremely interesting and relevant and I have some comments/remarks concerning the results, the presented analysis and framing of the results presented here in the view of earlier studies.

We truly appreciate Dr. Peters's recognition of the uniqueness of our manuscript and his interest in and comments on our manuscript. We provide our point-by-point response in the following.

Main point: In my view, the discussion of the differences between the two aerosol modules and their impact on the results warrants a bit more investigation/explanation. In Peters et al (2012), P12 in the following, we investigated the uncertainties of AIEs from shipping emissions related to the assumed emission particle size distribution and the total amount of fuel burnt. We found a significant impact of the assumed particle size distribution on the estimated AIEs: assuming all sulphuric compounds being assigned to the soluble Aitken mode at point of emission (as supported by various field observations) leads to significantly more negative AIEs than assigning them half and half to the soluble Accumulation and Coarse modes as was done in the standard Aerocom emission setup (cf P12). This is due to the substantially higher number of primary emitted soluble particles. This was further substantiated in the corrigendum to P12 (Peters et al., 2013), in which a bugfix in the aerosol module lead to an even higher number of emitted particles. My question here is: what size mode are the shipping emissions (and the DMS emissions) assigned to at the point of emission in MAM3 and MARC? This is not described in Section 2, but is a critical point. Compare to the detailed analysis from emission all the way to the resulting effects on cloud properties detailed in Peters et al. (2012, 2014), because in the end, AIEs are the end product of a long chain of processes calculated in various parameterizations with, most certainly, inherent uncertainties. Are there diagnostics of aerosol numbers per mode (see P12) available for the current study so as to investigate the differences between the two aerosol microphysics modules in more detail? This would also help in investigating the interplay with different assumed DMS emission levels, especially for the case of zero DMS emissions where the differences between the two parameterisations are largest. Can CCN diagnostics (see P12, Peters et al. 2014, P14 in the following) be provided to further investigate these points?

We added the following sentences at the end of Section 2.3 to further describe corresponding processes in MARC:

"Note that in MARC model, gas-phase sulfur compounds can be oxidized in both gaseous and aqueous phase to form sulfate that could enter aerosol phase in several pathways: (1) aerosol nucleation to form new nucleation mode sulfate aerosols; (2) condensation of gaseous sulfuric

acid on both external sulfate and carbonaceous aerosols (the latter specifically ages carbonaceous aerosols to form sulfate-carbonaceous aerosol mixtures); and (3) evaporation of cloud and rain drops that resuspends aqueous sulfate to accumulation mode sulfate aerosol (Kim et al., 2008; Grandey et al., 2018; Rothenberg et al., 2018)."

We have compared aerosol loadings of each aerosol type and CCN between MARC and MAM in another paper (Grandey et al., 2018), which is available (https://doi.org/10.5194/acp-2018-118).

To address this concern, we added the following discussion at the end of Section 3.5:
"To track down all the possible reasons for the differences in the ISE-induced CRE between the two aerosol schemes, more detailed analyses on a long chain of processes related to both aerosols and clouds are required, as done by Peters et al. (2014), which is out of the scope of this study and warrants more studies in the future. Interesting reader is referred to another paper of ours (Grandey et al., 2018) in addressing this aspect."

Minor points:

Lines 40-42: please also mention the Corrigendum to P12 – i.e. Peters et al 2013 - as the results from P12 suffered from a bug in the aerosol microphysics module. The results presented in Peters et al (2013) are thus more sound.

Done.

Lines 118 – 126: while investigating the effect of total sulfur content in bunker fuel is an important issue, please also mention uncertainties related to the total amount of fuel burnt (cf P12).

Now we have added the following sentence to address this concern:
"These numbers related to annual sulfur emissions are generally estimated based on the total amount of heavy fuel burnt by ships and the associated emission rates, whose uncertainties were addressed by Peters et al. (2012)."

Lines 170-172: see my above comment regarding an analysis of the causal chain from emissions -> AIEs.

We addressed this causal chain at the end of Section 3.5. Please also see our responses to the "Main point" above.

Lines 178-180: this reads like the increase in CWP leads to an increase in CDNC, but it should be the other way around (at least from the viewpoint of a parameterisation in which a causal connection of cause-and-effect has to be established by design)

In this manuscript, we presented our results in the following order: CRE<-CWP<-CDNC, which tracks back the reason that causes the changes in CRE. We believe that changes in CDNC result in changes in CWP, which in turn causes changes in CRE. Now we revised the sentence to reflect the cause-effect relation:

"The sulfate aerosols from shipping emissions are highly efficient cloud condensation nuclei (CCN) and thus can increase the CDNC, which in turn affects CWP."

Lines 181-182: this is obvious and is out of place at the end of this paragraph (compare to lines 30-34 in the Introduction)

This sentence is a short and brief summary of Section 3.2 and we revised it to clearly describe the cause-effect relation among ISE emission, CDNC, CWP, and CRE.

Lines 187-192: This is a very interesting paragraph, specifically because DMS emissions are natural and thus an integral part of the climate system and are most probably also included when tuning the TOA radiation balance of the model. Capturing the "correct" background (pre-industrial) aerosol distribution is an extremely difficult task, see e.g. Stevens et al 2017 for the case of developing an aerosol climatology, and is critical for estimating anthropogenic AIEs (as is very nicely shown in this paper). Coming to the point, leaving out DMS emissions results in a quite large TOA radiative imbalance in excess of -5 Wm-2 (Figure 9). Although the model is constrained by prescribed SSTs, this imbalance, which is much larger on local scales, could have an effect on the results.

We thank the reviewer for raising such as important point. We assume that the reviewer was referring to Figure 6, which shows the ISE-induced CRE with various DMS emissions. When DMS is turned off, the ISE-induced CRE can reach up to 6 W m$^{-2}$. We agree that this magnitude is very large and warrant further studies. The following sentences are added in this paragraph to emphasize on this point:

"It is worth noting that the ISE-induced CRE can reach up to −6 W m$^{-2}$ when DMS emission is turned off, such as over NPO and NAO, which is a very large negative forcing even on the local scale. Since there are no comparable values in the literature, this large negative forcing warrants a detailed evaluation in future studies using different climate models."

Lines 266-268: a very important point. Even more importantly, this calls for a reevaluation of aerosol and cloud microphysics parameterisations in general circulation models.

We have revised the sentence to emphasize on the point:
"From the perspective of simulation, this nonlinearity in aerosol activation strongly suggests a reevaluation of CRE induced by shipping and DMS emissions as well as a reevaluation of parameterizations of aerosols–cloud interactions in the general circulation models."

Lines 269-272: I completely agree. In Peters et al 2011, P11, we applied a specific sampling routine to observational data in order to sample for the effect of shipping emission on cloud properties in "pristine" oceanic areas, where "pristine" was mean with regards to anthropogenic emissions. Looking at the maps displayed in Figure 2, shipping emissions are trumped by DMS emissions in two of the regions sampled in P11: the SE Pacific and the mid-Indian Ocean region. However, the third region investigated in P11, the mid Atlantic, shows a significant contribution of shipping emissions compared to DMS. We also focused more on that region in P14 and

concluded that for "observational studies of AIEs, this highlights the ever so important and often discussed aspect of correctly defining the background ('pre-industrial') reference state against which to gauge the present-day observations." The present study thus very convincingly corroborates our conclusions drawn in 2014.

The importance of correctly defining the background aerosol level has been addressed in many studies, such as P14 mentioned by the reviewer. The uniqueness of conclusions in this manuscript is that the impact of DMS on shipping emission-induced CRE is understudied. Therefore, it has potentially important implication for future studies.

References:
Peters, K., J. Quaas, and H. Graßl (2011), A search for large  Rscale effects of ship ˇ emissions on clouds and radiation in satellite data, J. Geophys. Res., 116, D24205, doi: 10.1029/2011JD016531.

Peters, K., Stier, P., Quaas, J., and Graßl, H.: Aerosol indirect effects from shipping emissions: sensitivity studies with the global aerosol-climate model ECHAM-HAM, Atmos. Chem. Phys., 12, 5985-6007, https://doi.org/10.5194/acp-12-5985-2012, 2012

Peters, K., Stier, P., Quaas, J., and Graßl, H.: Corrigendum to "Aerosol indirect effects from shipping emissions: sensitivity studies with the global aerosol-climate model ECHAM-HAM" published in Atmos. Chem. Phys., 12, 5985–6007, 2012, Atmos. Chem. Phys., 13, 6429-6430, https://doi.org/10.5194/acp-13-6429-2013, 2013.

Peters, K., Quaas, J., Stier, P. and Graßl, H. :Processes limiting the emergence of detectable aerosol indirect effects on tropical warm clouds in global aerosol-climate model and satellite data, Tellus B: Chemical and Physical Meteorology, 66:1, DOI: 10.3402/tellusb.v66.24054, 2014.

Stevens, B., Fiedler, S., Kinne, S., Peters, K., Rast, S., Müsse, J., Smith, S. J., and Mauritsen, T.: MACv2-SP: a parameterization of anthropogenic aerosol optical properties and an associated Twomey effect for use in CMIP6, Geosci. Model Dev., 10, 433-452, https://doi.org/10.5194/gmd-10-433-2017, 2017.

We also thank the reviewer for providing these references.

**References**

Grandey, B. S., Rothenberg, D., Avramov, A., Jin, Q., Lee, H.-H., Liu, X., Lu, Z., Albani, S., and Wang, C.: Effective radiative forcing in the aerosol–climate model CAM5.3-MARC-ARG, *Atmospheric Chemistry and Physics Discussions*, doi:10.5194/acp-2018-118, 2018.

Kim, D., Wang, C., Ekman, A. M. L., Barth, M. C., and Rasch, P. J.: Distribution and direct radiative forcing of carbonaceous and sulfate aerosols in an interactive size-resolving aerosol-climate model, J Geophys Res-Atmos, 113, 2008.

Peters, K., Stier, P., Quaas, J., and Grassl, H.: Aerosol indirect effects from shipping emissions: sensitivity studies with the global aerosol-climate model ECHAM-HAM, Atmos Chem Phys, 12, 5985-6007, 2012.

Peters, K., Quaas, J., Stier, P., and Graßl, H.: Processes limiting the emergence of detectable aerosol indirect effects on tropical warm clouds in global aerosol-climate model and satellite data, Tellus B: Chemical and Physical Meteorology, 66, 24054, 10.3402/tellusb.v66.24054, 2014.

Pringle, K. J., Tost, H., Pozzer, A., Pöschl, U., and Lelieveld, J.: Global distribution of the effective aerosol hygroscopicity parameter for CCN activation, Atmos Chem Phys, 10, 5241-5255, 10.5194/acp-10-5241-2010, 2010.

Rothenberg, D., Avramov, A., and Wang, C.: On the representation of aerosol activation and its influence on model-derived estimates of the aerosol indirect effect, Atmos Chem Phys, 18, 7961-7983, 10.5194/acp-18-7961-2018, 2018.